# Quantitative Analysis and Stability Study on Iridoid Glycosides from Seed Meal of *Eucommia ulmoides* Oliver

**DOI:** 10.3390/molecules27185924

**Published:** 2022-09-12

**Authors:** Lulu Ma, Ning Meng, Benyu Liu, Changjian Wang, Xin Chai, Shan Huang, Huijuan Yu, Yuefei Wang

**Affiliations:** 1State Key Laboratory of Component-Based Chinese Medicine, Tianjin Key Laboratory of TCM Chemistry and Analysis, Tianjin University of Traditional Chinese Medicine, Tianjin 301617, China; 2State Key Laboratory of Component-Based Chinese Medicine, Haihe Laboratory of Modern Chinese Medicine, Tianjin University of Traditional Chinese Medicine, Tianjin 301617, China; 3Department of Pharmacy, Qingdao University of Science & Technology, Qingdao 266000, China

**Keywords:** seed meal of *Eucommia ulmoides* Oliver, anti-inflammatory, quality analysis, stability

## Abstract

As a traditional Chinese medicine, *Eucommia ulmoides* Oliver (*E. ulmoides* Oliv.) is an important medicinal plant, and its barks, male flowers, leaves, and fruits have high value of utilization. The seed meal of *E. ulmoides* Oliv. is the waste residue produced after oil extraction from seeds of *E. ulmoides* Oliv. Though the seed meal of *E. ulmoides* Oliv. is an ideal feed additive, its medicinal value is far from being developed and utilized. We identified six natural iridoid compounds from the seed meal of *E. ulmoides* Oliv., namely geniposidic acid (GPA), scyphiphin D (SD), ulmoidoside A (UA), ulmoidoside B (UB), ulmoidoside C (UC), and ulmoidoside D (UD). Six natural iridoid compounds were validated to have anti-inflammatory activities. Hence, six compounds were quantified at the optimum extracting conditions in the seed meal of *E. ulmoides* Oliv. by an established ultra-performance liquid chromatography (UPLC) method. Some interesting conversion phenomena of six tested compounds were uncovered by a systematic study of stability performed under different temperatures and pH levels. GPA was certified to be stable. SD, UA, and UC were only hydrolyzed under strong alkaline solution. UB and UD were affected by high temperature, alkaline, and strong acid conditions. Our findings reveal the active compounds and explore the quantitative analysis of the tested compounds, contributing to rational utilization for the seeds residues of *E. ulmoides* Oliv.

## 1. Introduction

*Eucommia ulmoides* Oliver (*E. ulmoides* Oliv.) is a Chinese medicinal material (CMM), endemic to China, which has been extensively employed for nearly two thousand years owing to its treatment of various diseases, such as lumbar pain, knee pain, osteoporosis, hepatoprotection, paralysis, etc. [1,2,3]. The barks and leaves of *E. ulmoides* Oliv. have been effectively used [3,4,5,6]. Moreover, the fruits of *E. ulmoides* Oliv. have been paid more attention to. Importantly, the gum of the fruits was proved to be the highest abundant, which can reach 12–18%, and always be employed in various fields, including aerospace, national defense, healthcare, transportation, sports, and construction [7,8,9,10]. Moreover, the oil content of seeds of *E. ulmoides* Oliv. is about 30%, which can be used for food, medicine, and industry [11,12]. However, the seed meal of *E. ulmoides* Oliv. is an ideal feed additive as the by-product after the oil is extracted and rich in protein, sugars, mineral elements, vitamins, and bioactive substances, such as aucubin, geniposide, geniposide acid, flavonoids, etc. [13], which was not fully utilized. Consequently, it is of great academic value and economic significance to study the chemical compounds, which may discover the active compounds and conduce to the improvement of quality analysis for seed meal of *E. ulmoides* Oliv.

At present, some researchers have isolated and identified iridoid glycosides, flavonoid glycosides, and organic acids from seeds of *E. ulmoides* Oliv. or seed meal. In particular, Tang et al. isolated seven iridoid glycosides compounds, including aucubin, bartsioside, linaride, geniposidic acid (GPA), scyphiphin D (SD), ulmoidoside A (UA), and ulmoidoside B (UB), from seed meal of *E. ulmoides* Oliv. and verified their anti-inflammatory activities [14]. It is widely known that the research on the quantitative analysis of bioactive compounds is beneficial to controlling the quality. Moreover, the possible transformation behaviors will happen for SD, UA, UB, etc., which are formed by GPA as the basic unit. However, minimal research on quantitative analysis and stability studies of compounds with anti-inflammatory activities from seed meal of *E. ulmoides* Oliv. have been reported [14,15,16,17]. There is no doubt that a systematic understanding of the stability of bioactive compounds would contribute to improvements in manufacturing and controlling the quality of CMMs [18,19,20]. Meanwhile, it is also vital to design the suitable sample packaging and choose storage conditions for products.

In this study, six natural iridoid compounds were extracted and identified from seed meal of *E. ulmoides* Oliv., including geniposidic acid (GPA, monomer), scyphiphin D (SD, dimer), ulmoidoside A (UA, trimer), ulmoidoside B (UB, trimer), ulmoidoside C (UC, tetramer), and ulmoidoside D (UD, tetramer). We have certified that six natural iridoid compounds have anti-inflammatory activities by evaluating nitrite levels. By the optimum extracting conditions optimized with a simple and visual model of “spider-web”, six tested compounds were simultaneously quantified in 5 min by ultra-performance liquid chromatography coupled with a photo-diode array (UPLC-PDA). More importantly, a systematic stability study of six tested compounds was performed under different temperatures and pH levels, resulting in the elucidation of the transformation behaviors for bioactive compounds.

## 2. Results and Discussion

### 2.1. Evaluation of Anti-Inflammatory Activity for Six Compounds from Seed Meal of E. ulmoides Oliv.

Six natural iridoids were extracted by 50% methanol aqueous solution from seed meal of *E. ulmoides* Oliv., including GPA as the basic unit, SD as dimer, UA and UB as trimers, and UC and UD as tetramers. Their structures were confirmed by NMR spectrometer [21,22], which are shown in Appendix A Appendix A.

Six natural iridoid glucosides were evaluated for their inhibitory effects on nitrite production in the lipopolysaccharide (LPS)-stimulated RAW 264.7 cells. Dexamethasone (DXMS) was used as positive control. By employing cell counting kit-8 (CCK8), effects of the tested compounds on cell viabilities of RAW 264.7 cells were assessed. As shown in Figure 1A, six tested compounds exhibited no significant cytotoxicity in the tested concentrations from 2.5 to 40 μM (cell viability >94.15%). As shown in Figure 1B, six tested compounds had good inhibitory effects on nitrite production in a concentration-dependent manner. Among the tested compounds, GPA, SD, UA, and UB have been proved to have anti-inflammatory activities, which is consistent with a previous report [14]. UC and UD were first reported to have anti-inflammatory activities in this study. Generally, it suggests that the anti-inflammatory activities of the tested compounds increased along with the increased amounts of GPA basic units of compounds, with the exception of GPA. UA and UC had weaker anti-inflammatory activities than their ethyl esterified products, UB and UD, respectively. As a tetramer esterified structure, UD had strong anti-inflammatory activity.

### 2.2. Optimization of Extraction Conditions for Six Tested Compounds

As proved in our study, six iridoid glucosides from the seed meal of *E. ulmoides* Oliv. showed anti-inflammatory activities. Aiming at establishment of quantitative method, we employed these compounds to evaluate the quality of the seed meal of *E. ulmoides* Oliv. In order to extract six iridoid glucosides from the seed meal of *E. ulmoides* Oliv. more efficiently, the “spider-web” mode, a simple and efficient method, was used to display the optimum extraction conditions, according to the multivariate valuation methods provided by our group [23,24].

To express this concisely, in the following Equation (1), $C_{m-k}$ represents the content of the tested compounds, and divides by corresponding maximum value ($C_{k}^{\prime }\left(\max\right)$) to obtain $\;E_{m-k}$. *k* stands for the tested compounds (GPA, SD, UA, UC, UB, and UD), and *m* is denoted as different extraction conditions (different solvents, solid–liquid ratios, and ultrasonic time).
(1)$$E_{m-k}=C_{m-k}/C_{k}^{\prime }\left(\max\right)$$

By taking optimization of extraction solvent as an example (Figure 2A), GPA, SD, UA, UC, UB, and UD were employed to establish six dimensions of the “spider-web” mode ($p_{i}$) by ultrasonic extraction at 40 °C. The shaded area of the “spider-web” mode was calculated by Equation (2). The optimal extraction conditions were screened according to the shaded area. The angle between the adjacent variables was denoted as *α* (*α* = 360°/*n*, *n* = 6), respectively [23,24,25].
(2)$$S=\frac{1}{2}\sin\alpha \left(\overset{n-1}{\underset{\dot{l}=1}{\sum }}p_{i}\cdot p_{i+1}+p_{n}\cdot p_{1}\right)$$

Based on the systematic research, we testified that 60% methanol aqueous solution was the successful solvent to extract the compounds of interest with a “spider-web” area at 2.60, after an exhaustive screening of different solvents (15%, 30%, 45%, 60%, 75%, and 90% methanol aqueous solution). On the basis of the focused extraction solvent, the solid–liquid ratio (1:50, 1:125, and 1:250) and ultrasonic time (15 min, 30 min, and 45 min) were investigated as well (Figure 2B). We confirmed that the optimal solid–liquid ratio to extract the compounds of interest was 1:125 with a “spider-web” area at 2.59, and the optimum ultrasonic time was 30 min with a “spider-web” area at 2.59. Generally, the sample powder was ultrasonically extracted by 60% methanol aqueous solution for 30 min at a ratio of 1:125 (material: solvent) at 40 °C.

**Figure 2 molecules-27-05924-f002:**
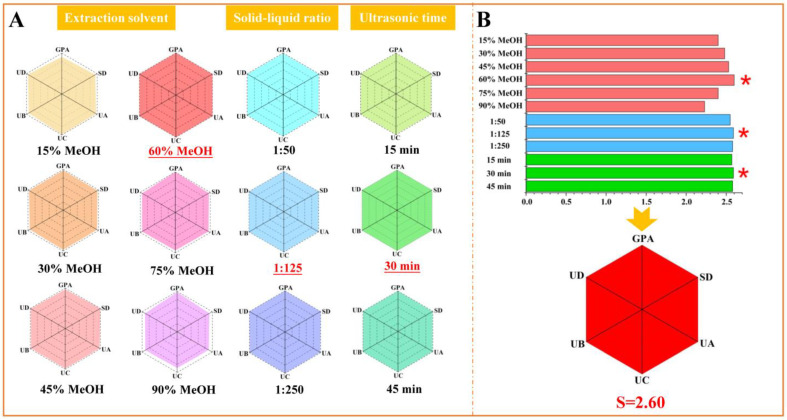
Optimization of the extracting method for seed meal of *E. ulmoides* Oliv. using the “spider-web” mode, including extraction solvent, solid-liquid ratio, and ultrasonic time. * The optimal values.

### 2.3. Methodological Validation of the Quantitative Analysis and Quantificaiton of Six Tested Compounds in Seed Meal of E. ulmoides Oliv.

The developed UPLC-PDA method was validated for quantitative analysis of the tested compounds. The specific results were summarized in Table 1. Excellent linear relationship of all six tested compounds was reached in the detected range. The limit of detection (LOD) and limit of quantitation (LOQ) values were less than 0.102 and 0.322 μg/mL, respectively. Intra- and inter-day precisions were achieved to the ideal level, and stability with all RSDs was below 1.5%. What is more, the average recoveries of the tested compounds were 90.83–115.30% with all RSDs below 2.34%. Therefore, the established UPLC-PDA method was reliable and appropriate for simultaneously determining six compounds from the seed meal of *E. ulmoides* Oliv.

The representative chromatograms of the mixed reference compounds and the sample solution of the seed meal of *E. ulmoides* Oliv. are shown in Figure 3A,B. Through the quantitative analysis of six iridoid glycosides in samples from different origins, the contents of the tested compounds were greatly fluctuated. As trimers (UA and UB), UA was proved to be the most abundant, whose content in the seed meal of *E. ulmoides* Oliv. is in the range of 24.45–31.85 mg/g, while UB’s content is 3.85–10.42 mg/g lower than UA. UC and UD are classified into tetramers, whose contents are in the range of 15.78–22.40 mg/g and 6.220–14.11 mg/g, respectively. Compared to other compounds, the content of GPA (0.6783–2.580 mg/g) as a monomer and SD (1.653–2.014 mg/g) as a dimer were witnessed to be significantly lower. In order to intuitively and clearly display the distribution of the contents of interesting iridoid glucosides, a heatmap was employed in this study (Figure 3C). We normalized the data and expressed the quantified results by relative contents. The same trend was witnessed. The redder the color is, the higher the content is, whereas the bluer the color is, the lower the content is.

### 2.4. Effects of Temperatures and pH on Stability of GPA, SD, UA, UC, UB, and UD

GPA, SD, UA, UC, UB, and UD are the representative iridoid glucosides existing in seed meal of *E. ulmoides* Oliv. As a basic unit, GPA is connected by ester bonds to produce SD (dimer), UA (trimer), and UC (tetramer). Moreover, UA and UC undergo ethyl esterification to generate UB and UD. These compounds characterized with ester bonds are susceptible to the variation of temperature and pH environment, which should be paid more attention to.

Shown in Figure 4A1 are the representative chromatograms of the tested sample solution exposed to different temperatures. In Figure 4A2, it is obviously displayed that UC is inert to the increasing temperatures (20, 40, 60, and 80 °C) as a tetramer. As shown in Figure 4B1–B4, UC was proved to be stable at pH ≤ 8 and gradually degraded into UA, SD, and GPA at pH ≥ 10. Especially, at pH 12, UC was fully converted into GPA after 18 h. The degraded pathways were proposed in Figure 4C. Likewise, displayed in Appendix A, UA and SD were also employed to perform a stability study, which went through the similar degradation pathways by hydrolysis of ester bonds under pH ≥ 10. UA and SD turned out stability under the temperatures (20–80 °C) and pH (≤8). In Appendix A Appendix A, GPA as a monomer was certified to be the most stable in the tested conditions.

As an ethyl esterified tetramer, in the tested range of 20 °C to 80 °C, UD was relatively stable (Figure 5A1,A2). As shown in Figure 5B1–B6, at pH 4 and 6, UD was witnessed to be stable; at pH 2 and 8, UD was slowly converted into UC; at pH 10, UD was almost hydrolyzed into UC at 6 h, and slowly converted into UA, SD, and GPA; at pH 12, UD was almost degraded into GPA at 30 h. It is obvious that along with the increased pH, esterification bonds were completely hydrolyzed to produce GPA. The proposed degradation pathways of UD were listed in Figure 5C. As an ethyl esterified trimer, the degradation pathways of UB were similar to that of UD (Appendix A Appendix A).

In order to intuitively and clearly display the stability of six iridoid glucosides under the conditions of different temperatures and pH levels, we employed degradation degree (*P*) to express instability degree of the tested compounds. In the following Equation (3), *C_k_* represents the initial concentration of the tested compounds at 0 h, $C_{k}^{\prime }$ stands for the concentration of the tested compounds in the exposed conditions at 30 h, and *k* stands for the different tested compounds. As shown in Figure 6, the redder the color is, the greater the degradation degree of the tested compound is.
(3)$$P=\frac{C_{k}-C_{k}^{\prime }}{C_{k}}\times 100\%$$

## 3. Materials and Methods

### 3.1. Reagents and Materials

Acetonitrile and methanol with HPLC-grade were purchased from Sigma-Aldrich Crop. (St. Louis, MO, USA). Formic acid was bought from Shanghai Aladdin Bio-Chem Technology Co., Ltd. (Shanghai, China). Water used in this study was purchased from Guangzhou Watson’s Food & Beverage Co., Ltd. (Guangzhou, China). Seed meal of *E. ulmoides* Oliv. was obtained from Shandong Beilong Eucommia Bioengineering Co., Ltd. (Shandong, China). The reference compounds (GPA, SD, UA, UC, UB, and UD) were purified in our lab and identified by NMR spectrometer, whose purities above 95% (*w*/*w*) were determined by UPLC analysis.

Mouse macrophage RAW 264.7 cells were obtained from National Collection of Authenticated Cell Cultures (Shanghai, China). Dulbecco’s modified Eagle medium (DMEM) and fetal bovine serum (FBS) were purchased from Gibco BRL Life Technologies, Inc. (Grand Island, NY, USA). Penicillin-streptomycin (PS) was obtained from Hyclone (South Logan, UT, USA). Griess reagent was bought from Beyotime Biotechnology (Shanghai, China). Lipopolysaccharide (LPS) was obtained from Sigma-Aldrich Crop. (St. Louis, MO, USA). Dimethyl sulfoxide (DMSO) was purchased from Beijing Solarbio Science & Technology Co., Ltd. (Beijing, China). Cell counting kit-8 (CCK8) was bought from MedChem Express (Monmouth Junction, NJ, USA).

According to the guidelines released by Chinese pharmacopoeia [26], the pH 2 buffer solution was prepared by phosphoric acid and disodium hydrogen phosphate, the buffer solutions at pH 4 and 6 were obtained by dissolving acetic acid and sodium acetate, the buffer solution at pH 8 was formed with dipotassium hydrogen phosphate and potassium dihydrogen phosphate, and the pH 10 buffer solution was made of borax and sodium carbonate. The buffer system at pH 12 was composed of borax, sodium carbonate, and sodium hydroxide. In our initial trial, the above buffer systems at pH 4, 6, 10, and 12 were unsuitable for the stability study of GPA. For the study of GPA stability, the buffer solutions at pH 4 and 6 were prepared by phosphoric acid and disodium hydrogen phosphate [27], the buffer solutions at pH 10 was formed with dipotassium hydrogen phosphate and sodium hydroxide, respectively [26], and the buffer solutions at 12 were formed with dipotassium hydrogen phosphate and sodium hydroxide.

### 3.2. Cell Viability Assay

The cell viability exposed to the tested compounds was estimated by CCK-8 assay. The mouse macrophage cell line RAW 264.7 was cultured in high glucose DMEM and supplemented with 10% FBS and 1% PS (standard growth medium) under a humidified atmosphere containing 5% CO_2_ and 95% air at 37 °C. The RAW 264.7 cells were seeded into 96-well cell culture plates by adding 100 μL at a density of 5 × 10^4^ cells/mL and incubated for 24 h. The supernatant was removed. The tested compound’s solutions (100 μL) at 2.5, 5, 10, 20, and 40 μM were individually employed to pretreat cells for 24 h, which were, respectively, prepared by DMSO to obtain stock solutions at 50 mM and diluted with standard growth medium. Control group cells were treated with 100 μL standard growth medium without the tested compounds. Then, the supernatant was replaced with 100 μL fresh DMEM. The cells were continuously incubated by adding 10 μL CCK-8 solution at 37 °C for 2 h. The absorbance values of each well at 450 nm were measured by a microplate reader under dark condition at 37 °C. Unstimulated RAW 264.7 cells were conducted as control, whose absorbance value was defined as 100%. All assays were carried out in triplicate.

### 3.3. Measurement of Nitrite Levels

The anti-inflammatory effects of the tested compounds on RAW 264.7 cells induced by LPS were determined by nitrite levels according to the Griess assay [28]. The mouse macrophage cell line RAW 264.7 was cultured in high glucose DMEM supplemented with 10% FBS and 1% PS (standard growth medium) in a humidified atmosphere containing 5% CO_2_ and 95% air at 37 °C. RAW 264.7 cells were cultured in 96-well plates by adding 100 μL at a density of 5 × 10^5^ cells/mL and incubated in standard culture conditions for 24 h. The experiments were divided into four groups: control group, LPS (1 μg/mL) group, DXMS group, and experimental group. Then, the supernatant was removed. Control group cells were treated with 100 μL standard growth medium without the tested compounds. For the anti-inflammatory activity study, six tested compounds were, respectively, dissolved in DMSO to reach concentration at 50 mM and diluted with standard growth medium to obtain the tested compound’s solutions at 2.5, 5, 10, 20, and 40 μM, which (100 μL) were separately used to treat cells co-stimulated with 1 µg/mL LPS in experimental groups for 24 h. Dexamethasone (DXMS, 10 μg/mL) was used as positive control. The supernatant (50 μL) from each well was transferred into a 96-well plate. Then, Griess reagent I (50 μL) and Griess reagent II (50 μL) were added successively and absorbance was measured at 540 nm to calculate nitrite concentration by the established standard curve of sodium nitrite (*y* = (*x* − 0.034)/0.0086) using a microplate reader. All assays were carried out in triplicate.

### 3.4. Preparation of Standard Solution

Six reference compounds, GPA, SD, UA, UC, UB, and UD, were accurately weighed, dissolved, and diluted with 10% methanol aqueous solution to obtain stock solutions separately at final concentrations at 1 mg/mL, 0.31 mg/ mL, 2.50 mg/mL, 2.50 mg/mL, 1 mg/mL, and 2.50 mg/mL, which were applied to prepare mixed standard solution at concentrations of 65.33 μg/mL for GPA, 60.02 μg/mL for SD, 513.1 μg/mL for UA, 330.7 μg/mL for UC, 203.8 μg/mL for UB, and 208.9 μg/mL for UD, respectively. Then, six different concentrations were obtained by serially diluting the mixed reference solution with 10% methanol aqueous solution (*v*/*v*) for establishing calibration curves.

### 3.5. Preparation of Sample Solution

An accurately weighed sample powder (0.2 g) was added by about 20 mL of 60% methanol aqueous solution in a volumetric flask (25 mL). Then, it was performed by ultrasonic extraction at 40 °C for 30 min and cooled to the ambient temperature. The sample solution was diluted to the mark with 60% methanol aqueous solution and centrifuged at 12,700 rpm for 20 min. The supernatant was diluted with 10% methanol aqueous solution at a volume ratio of 1:1 and injected into UPLC-PDA for analysis.

### 3.6. UPLC-PDA Analysis

UPLC analysis was undertaken on an ACQUITY UPLC BEH Shield RP18 column (2.1 mm × 100 mm, 1.7 μm) at 40 °C by using a Waters ACQUITY UPLC system. The mobile phase, 0.1% formic acid solution (A), and acetonitrile (B) were eluted in a gradient program of 5–30% B in 0–4 min and 30–35% B in 4–5 min at a flow rate of 0.3 mL/min. During the stability study of this experiment, the gradient elution implemented was listed below: 5–35% B in 0–2 min, 35–45% B in 2–3 min, and 45–55% B in 3–3.5 min. The detection wavelength was monitored at 237 nm. Injection volume was 2 μL.

### 3.7. Methodological Validation

The quantitative analysis method established in this study was validated in the following aspects: linearity, LOD, LOQ, precision (intra- and inter-day), stability, repeatability, and recovery testing. Calibration curves were established based on the peak area (*y*) against the corresponding six different concentrations (*x*) of each compound. LOD and LOQ were detected based on signal-to-noise ratio at about 3 and 10, respectively. Intra- and inter-day precision were assessed by testing the same sample solution six times on the same day and three consecutive days, separately. To test the stability, the sample solution was injected repeatedly at 0, 2, 4, 6, 8, 10, and 12 h, respectively. Repeatability was verified by processing and analyzing six sample solutions. In order to evaluate the accuracy of the method, recovery testing was performed by adding a known amount of the mixed standard solution to 0.1 g sample powder, which was further prepared by the established method.

### 3.8. Stability Study of the Tested Compounds under Different Temperatures and pH Levels

GPA, SD, UA, UC, UB, and UD were dissolved in 200 mL of water to obtain sample solutions at final concentrations at about 40, 24, 13, 11, 14, and 12 μM, respectively. A 2 mL aliquot of sample solution for each compound was incubated at the different temperatures (20, 40, 60, and 80 °C) for 30 h. The solution was diluted with methanol at a volume ratio of 1:1 and analyzed according to the above-mentioned UPLC method to record concentrations of the tested compounds per 3 h.

GPA, SD, UA, UC, UB, and UD were dissolved in 10 mL of water. A 1.5 mL aliquot of sample solution was mixed with 48.5 mL of buffer solutions with different pH (2, 4, 6, 8, 10, and 12) to obtain sample solutions at final concentrations at about 40, 24, 13, 11, 14, and 12 µM, respectively. They were incubated at 40 °C for 30 h. The solution was diluted with methanol at a volume ratio of 1:1 and analyzed according to the above-mentioned UPLC method to record concentrations of the tested compounds per 3 h. The stability test of each compound was carried out three times.

### 3.9. Data Analysis

The “spider-web” mode, heat map, and line graphs were performed by origin 2019 software (OriginLab Ltd., Northampton, MA, USA) and Adobe Illustrator 2020 (Adobe Systems Incorporated, San Jose, CA, USA). The bar graph was plotted using GraphPad Prism 9.0 software (Graphpad software Inc., La Jolla, CA, USA).

## 4. Conclusions

Six iridoid glucosides, GPA, SD, UA, UC, UB, and UD, were isolated and identified from the seed meal of *E. ulmoides* Oliv. And proven to have anti-inflammatory activities. Confirmed by the well-established UPLC-PDA method, quantitative analysis was simultaneously performed for the tested compounds in the seed meal of *E. ulmoides* Oliv. from different origins. The contents of the tested compounds were greatly fluctuated, from which UA was shown to be the most abundant. More importantly, some interesting conversion phenomena of six tested compounds were uncovered. By hydrolysis of ester bonds under certain conditions, the compounds with more GPA units were decomposed into ones with fewer GPA units. It suggests that the divergence of contents for the tested compounds may arise from hydrolysis of ester bonds. The results contribute to improvements in controlling the quality of the seed meal of *E. ulmoides* Oliv. and may pave the way for a new form of recycling and harmless utilization for the seeds residue of *E. ulmoides* Oliv.

## Figures and Tables

**Figure 1 molecules-27-05924-f001:**
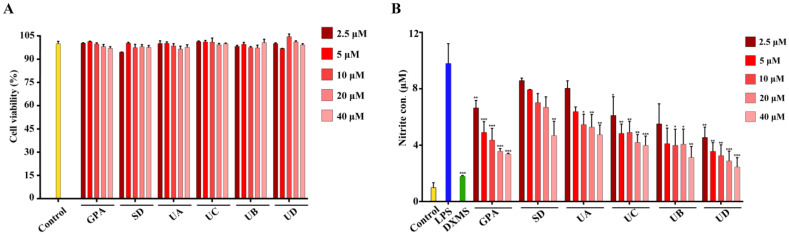
Cell viabilities of RAW 264.7 cells affected by six iridoid glucosides using cell counting kit-8 (CCK8) assay (**A**); The anti-inflammatory effects of the tested compounds on RAW 264.7 cells induced by lipopolysaccharide (LPS) by determining nitrite levels (**B**). * *p* < 0.05, ** *p* < 0.01, and *** *p* < 0.001 vs. LPS group (*n* = 3). (GPA, geniposidic acid; SD, scyphiphin D; UA, ulmoidoside A; UC, ulmoidoside C; UB, ulmoidoside B; and UD, ulmoidoside D).

**Figure 3 molecules-27-05924-f003:**
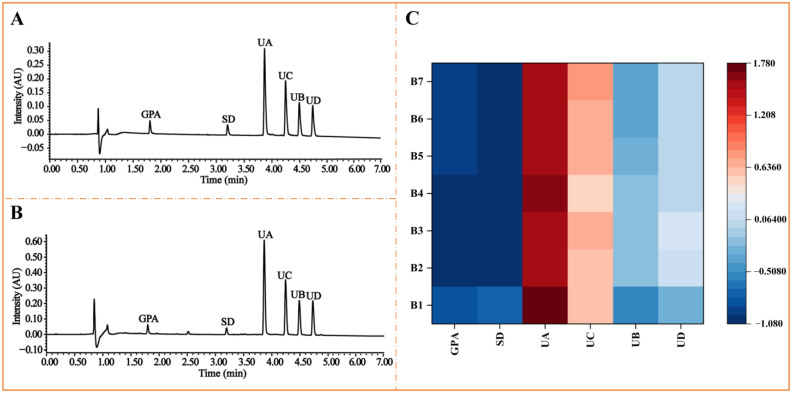
The representative UPLC chromatograms of the mixed reference compounds (**A**), and the sample solution of seed meal of *E. ulmoides* Oliv. (**B**); The heatmap of relative contents of the tested compounds in seed meal of *E. ulmoides* Oliv. from different origins (**C**).

**Figure 4 molecules-27-05924-f004:**
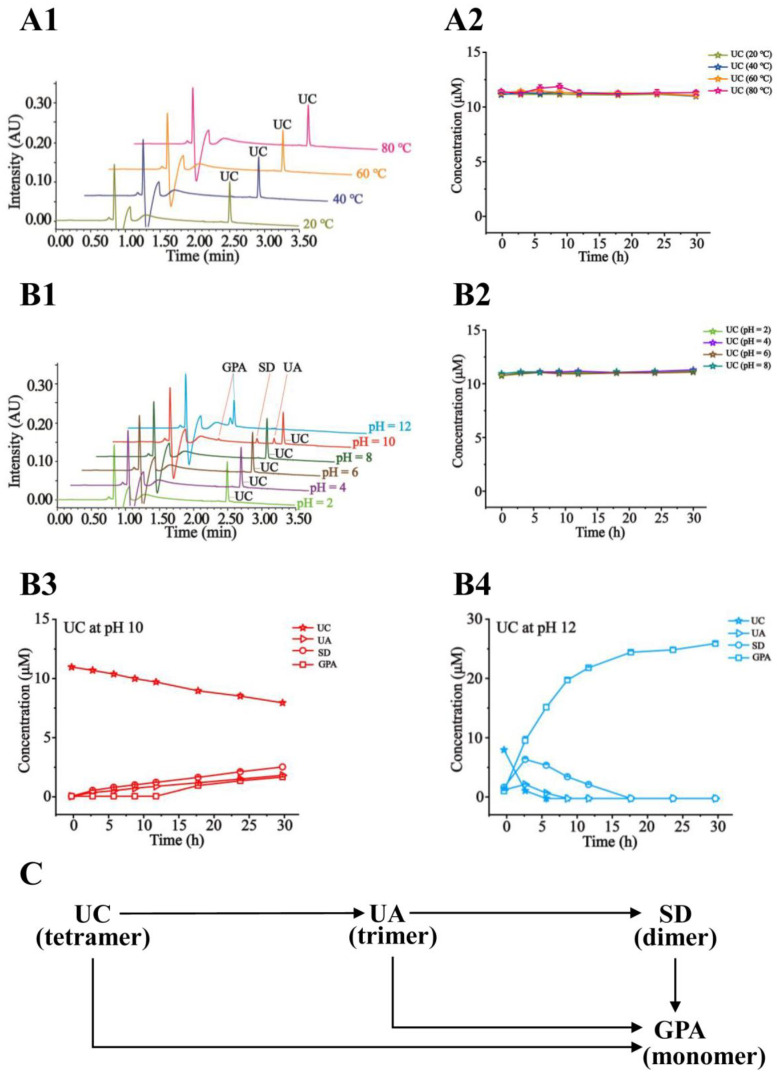
Representative chromatograms of UC in the tested solution exposed to different temperatures (**A1**) and pH levels (**B1**) after 24 h; Time–concentration curves of UC at different temperatures (**A2**) and pH 2~8 (**B2**); Time–concentration curves of UC and the degraded products at pH 10 (**B3**) and pH 12 (**B4**); The proposed degradation pathways of UC (**C**).

**Figure 5 molecules-27-05924-f005:**
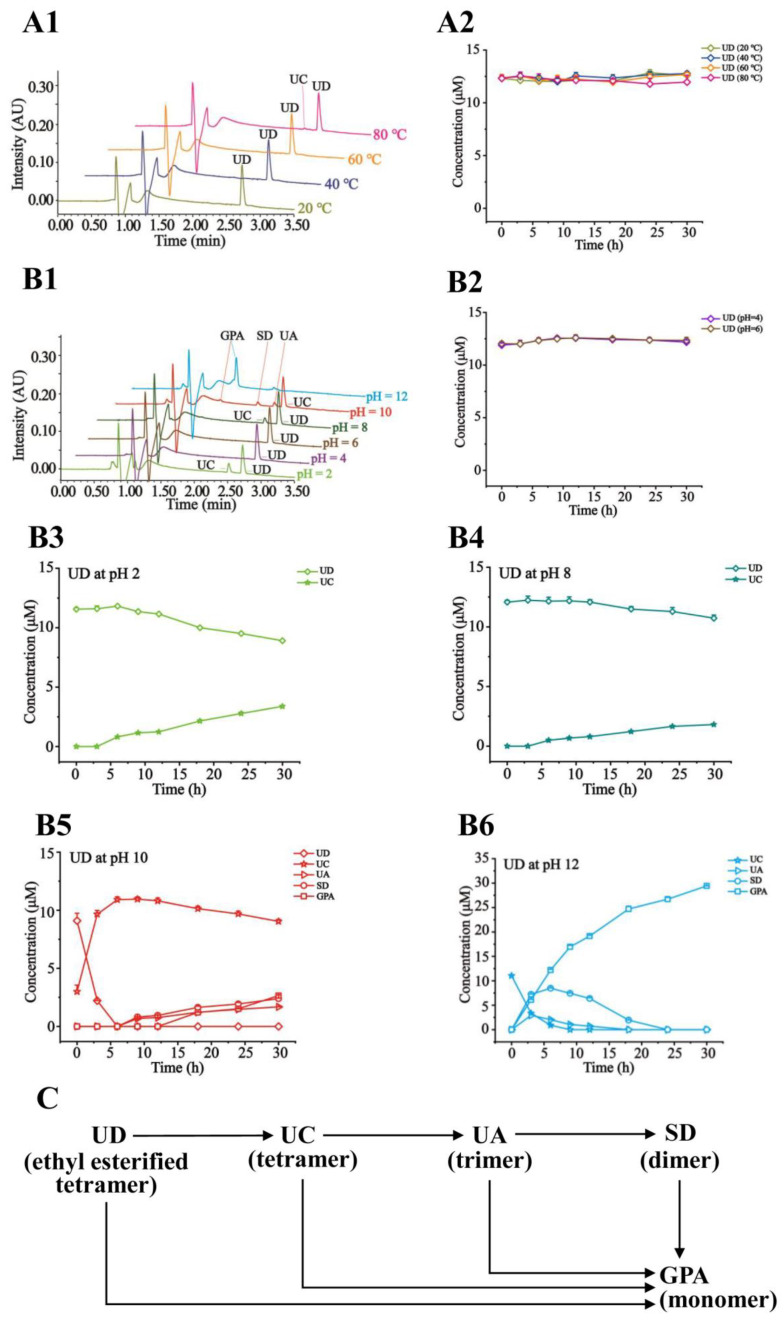
Representative chromatograms of UD in the tested solution exposed to different temperatures (**A1**) and pH levels (**B1**) after 24 h; Time–concentration curves of UD at different temperatures (**A2**) and pH 4~6 (**B2**); Time–concentration curves of UD and the degraded products at pH 2 (**B3**), pH 8 (**B4**), pH 10 (**B5**), and pH 12 (**B6**); The proposed degradation pathways of UD (**C**).

**Figure 6 molecules-27-05924-f006:**
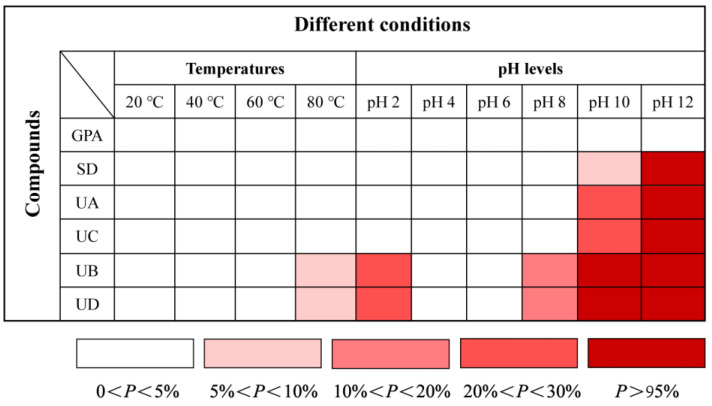
Illustration of stability degree for the tested compounds under the conditions of different temperatures and pH levels.

**Table 1 molecules-27-05924-t001:** Summary results of linear regression, LODs, LOQs, precision, repeatability, stability, and recovery for six tested compounds in seed meal of *E. ulmoides* Oliv.

Compounds	Linear Regression			LODs (μg/mL)	LOQs (μg/mL)	Precision (RSD, %)		Repeatability (*n* = 6, RSD, %)	Stability (*n* = 6, RSD, %)	Recovery (*n* = 6, Mean ± SD, %)
	Regression Equation	*r* ^2^	Linear Range (μg/mL)			Intra-Day (*n* = 6)	Inter-Day (*n* = 3)			
GPA	*y* = 9209.7 *x −* 631.79	0.9999	2.031–65.00	0.063	0.254	0.3	0.1	0.8	0.5	95.27 ± 1.40
SD	*y* = 8883.6 *x* + 3985.4	0.9996	1.876–60.02	0.059	0.234	0.5	0.9	0.3	0.3	90.84 ± 0.31
UA	*y* = 9343.4 *x* + 18,053.79	0.9996	16.02–512.5	0.063	0.250	0.2	0.9	0.1	0.2	96.35 ± 1.63
UC	*y* = 9802.2 *x −* 13,433.35	0.9999	10.31–330.0	0.081	0.322	0.2	1.0	0.5	0.2	115.3 ± 0.65
UB	*y* = 9785.1 *x −* 6293.28	0.9999	6.344–203.0	0.099	0.198	0.2	1.4	0.6	0.2	101.4 ± 1.02
UD	*y* = 9103.4 *x −* 8611.40	0.9999	6.516–208.5	0.102	0.204	0.1	1.5	0.6	0.2	106.5 ± 2.50

## Data Availability

Not applicable.

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
