# Peer review of "Quantitative Analysis and Stability Study on Iridoid Glycosides from Seed Meal of Eucommia ulmoides Oliver"

_molecules, 2022, doi:10.3390/molecules27185924_

Round 1

Reviewer 1 Report

The authors report about purification and identification of iridoid glycosides from seed meal of Eucommia ulmoides Oliver with anti-inflammatory activities.

The paper is well structured and well written.

There are some minor issues that the authors may want to consider before publication.

1.     It should be described in more detail in which solution the tested compounds were dissolved to determine the cell viability and the anti-inflammatory activity. Were these solutions analyzed for a control without the tested compounds?

2.     Please add section «Abbreviations».

3.     Line 52 ­– please write correct «Tang et al….»  instead «tang et al….».

Reviewer 2 Report

The article, Quantitative Analysis and Stability Study on Iridoid Glycosides from seed meal of Eucommia ulmoides Oliver is interesting but I see some points that need improvement.

The article appears to be fragmented. The introduction gives an impression that compounds will be isolated from the species Eucommia ulmoides Oliver. However, throughout the manuscript, I had another impression, including finding techniques that involve cell culture.

I believe that the article needs to have more connection between the topics. The introduction brings an idea of the theme and the results and discussion, another completely different.

- Figures need clearer captions and more description of the acronyms used.

Round 2

Reviewer 2 Report

The article can be accepted for publication.